# Structure and mechanism of the Mrp complex, an ancient cation/proton antiporter

**Julia Steiner, Leonid Sazanov\***

Institute of Science and Technology Austria, Klosterneuburg, Austria

**Abstract** Multiple resistance and pH adaptation (Mrp) antiporters are multi-subunit $Na^+$ (or $K^+$)/$H^+$ exchangers representing an ancestor of many essential redox-driven proton pumps, such as respiratory complex I. The mechanism of coupling between ion or electron transfer and proton translocation in this large protein family is unknown. Here, we present the structure of the Mrp complex from *Anoxybacillus flavithermus* solved by cryo-EM at 3.0 Å resolution. It is a dimer of seven-subunit protomers with 50 trans-membrane helices each. Surface charge distribution within each monomer is remarkably asymmetric, revealing probable proton and sodium translocation pathways. On the basis of the structure we propose a mechanism where the coupling between sodium and proton translocation is facilitated by a series of electrostatic interactions between a cation and key charged residues. This mechanism is likely to be applicable to the entire family of redox proton pumps, where electron transfer to substrates replaces cation movements.

## Introduction

The $Na^+$/$H^+$ antiporters are widely distributed secondary active transporters that use the proton motive force to efflux intracellular sodium ions (*Ito et al., 2017*). Several protein families catalyse this reaction and are mostly encoded by a single gene, such as the NHE family in eukaryotes and the NhaA family in bacteria (*Krulwich et al., 2011*). Mrp antiporters are unique as they usually consist of seven subunits (MrpABCDEFG) encoded in a single operon (*Figure 1—figure supplement 1a*). Because of the operon's distinctive properties, Mrp antiporters have been classified in a separate category, cation:proton antiporter-3 (CPA3), in the transporter classification system (*Saier et al., 2016*). They support intracellular pH homeostasis and $Na^+$ efflux in alkali- and halophilic microorganisms and are essential for their survival in challenging environments (*Hamamoto et al., 1994*; *Ito et al., 1999*). At high pH, the pH component of the proton motive force is inverted from normal, and so a substantial $\Delta\psi$ component (electric potential) is crucial to drive proton translocation into the cell (*Krulwich et al., 2011*). Antiporters with roles in alkaline pH homeostasis must catalyse electrogenic antiport, in which the ratio of $H^+$ entering the cell in exchange for $Na^+$ moving out is unequal, enabling proton entry to be driven by the $\Delta\psi$. For example, the stoichiometry for *E. coli* NhaA is $2H^+$/$1Na^+$ (*Taglicht et al., 1993*). The exact value for Mrp has not been fully established experimentally due to challenges in purification of the intact complex (*Morino et al., 2014*) but is likely to be also about 2, consistent with its function (*Dzioba-Winogrodzki et al., 2009*). This raises the question as to why a so much more complicated protein assembly is needed to catalyse a similar reaction. Since under the extreme environmental conditions Mrp is essential for cell survival and cannot be replaced by single subunit antiporters (*Cheng et al., 2016*; *Xu et al., 2018*), one of the proposals is that the Mrp complex can support cell growth at very high pH due to its large surface area, where only a few available external protons can still be gathered for translocation into the cell (*Ito et al., 2017*).

**\*For correspondence:**
sazanov@ist.ac.at

**Competing interests:** The authors declare that no competing interests exist.

The largest two Mrp subunits, MrpA and MrpD, are homologous to each other, have 14 conserved trans-membrane (TM) helices, and are thought to participate in proton translocation (*Mathiesen and Hägerhäll, 2003*). Their homologues (called antiporter-like subunits, which we will abbreviate to APLS) are found in many proton-pumping protein complexes where they are present in one to three (and recently discovered four [*Chadwick et al., 2018*]) copies per complex, depending on the energy availability and needs of the organism (*Efremov and Sazanov, 2012*). These include bacterial (*Baradaran et al., 2013*; *Friedrich et al., 1995*) and mitochondrial respiratory complex I (*Fiedorczuk et al., 2016*; *Walker, 1992*), NDH (NADH dehydrogenase-like) complex from cyanobacteria (*Laughlin et al., 2019*; *Pan et al., 2020*; *Schuller et al., 2019*) and chloroplasts (*Sazanov et al., 1998*), Fpo (F420:methanophenazine oxidoreductase) complex from archaea (*Baumer et al., 2000*) as well as various membrane-bound hydrogenases (*Efremov and Sazanov, 2012*) including MBH (membrane-bound [NiFe]-hydrogenase) complex from archaea (*Yu et al., 2018*). These modern enzymes represent some of the largest membrane protein complexes known and are thought to have evolved from the unification of the membrane transporter Mrp-like module with the soluble NiFe-hydrogenase module, sometimes followed by the addition of an electron input module, such as the NAD-linked formate dehydrogenase in case of complex I (*Efremov and Sazanov, 2012*). The Mrp complex thus represents an ancient ancestor of diverse protein families and is thought to have been among the few membrane proteins present in the last common ancestor of prokaryotes (*Sousa et al., 2016*).

Structures of complex I (*Agip et al., 2018*; *Baradaran et al., 2013*; *Fiedorczuk et al., 2016*; *Zickermann et al., 2015*), NDH (*Schuller et al., 2019*) and MBH (*Yu et al., 2018*) complexes have been solved recently. These enzymes consist of two main domains – the Mrp-like membrane domain, responsible for proton translocation (or sodium in case of MBH) and an attached hydrophilic redox domain, responsible for electron transfer between substrates (e.g. NADH to quinone in case of complex I). Therefore, the electron transfer and proton translocation processes are separated by large distances (up to 200 Å) and how they are coupled to each other remains a mystery. Upon solving the first structures of complex I, we proposed that redox reactions may drive proton translocation via long-range conformational changes (*Baradaran et al., 2013*; *Efremov and Sazanov, 2011*). However, such changes have not been visualized till now despite significant efforts (*Parey et al., 2018*). Electrostatic interactions between the key charged residues have also been proposed to play an additional (*Efremov and Sazanov, 2011*) or main (*Kaila, 2018*; *Verkhovskaya and Bloch, 2013*) role in the mechanism.

The structure of the universal common ancestor of these enzymes, the Mrp complex, has been lacking so far. Clearly, it would be instrumental in resolving the coupling mechanism, which should have common principles for this huge group of protein families. Furthermore, inactivation of the Mrp complex strongly reduces pathogenicity of such problematic human pathogens as *S. aureus* and *P. aeruginosa* (*Krulwich et al., 2011*), presenting Mrp as a valuable drug target. To address these questions, we have determined the first, to our knowledge, atomic structure of the Mrp complex.

## Results

### Structure determination

The Mrp complex from *Anoxybacillus flavithermus* shows high sequence similarity to the well-characterised Mrp complexes from *Bacillus* sp (*Figure 4—figure supplements 3–4*). The His-tagged Mrp complex from *A. flavithermus* was recombinantly expressed in the antiporter-deficient *E. coli* strain KNabc (*Goldberg et al., 1987*). The complex showed high $Na^+$ (and to a lesser degree $K^+$)/$H^+$ antiport activity and purified to high homogeneity, predominantly as a dimer (*Figure 1—figure supplement 1b,f–j*). It showed higher apparent stability than the *Bacillus* complex (*Morino et al., 2014*). The protein tended to aggregate heavily in ice holes of cryo-EM grids. Therefore, we used grids coated with a very thin layer of continuous carbon, which resulted in a uniform particle distribution (*Figure 1—figure supplement 2*). However, the particles showed a strong preferred orientation, with the hydrophilic protein surface attached to the carbon. To compensate for the associated loss of information, data collection was performed with grids tilted (*Tan et al., 2017*) at 35°, resulting in excellent quality maps (*Figure 1—figure supplements 2–4*). The initial dataset was collected with

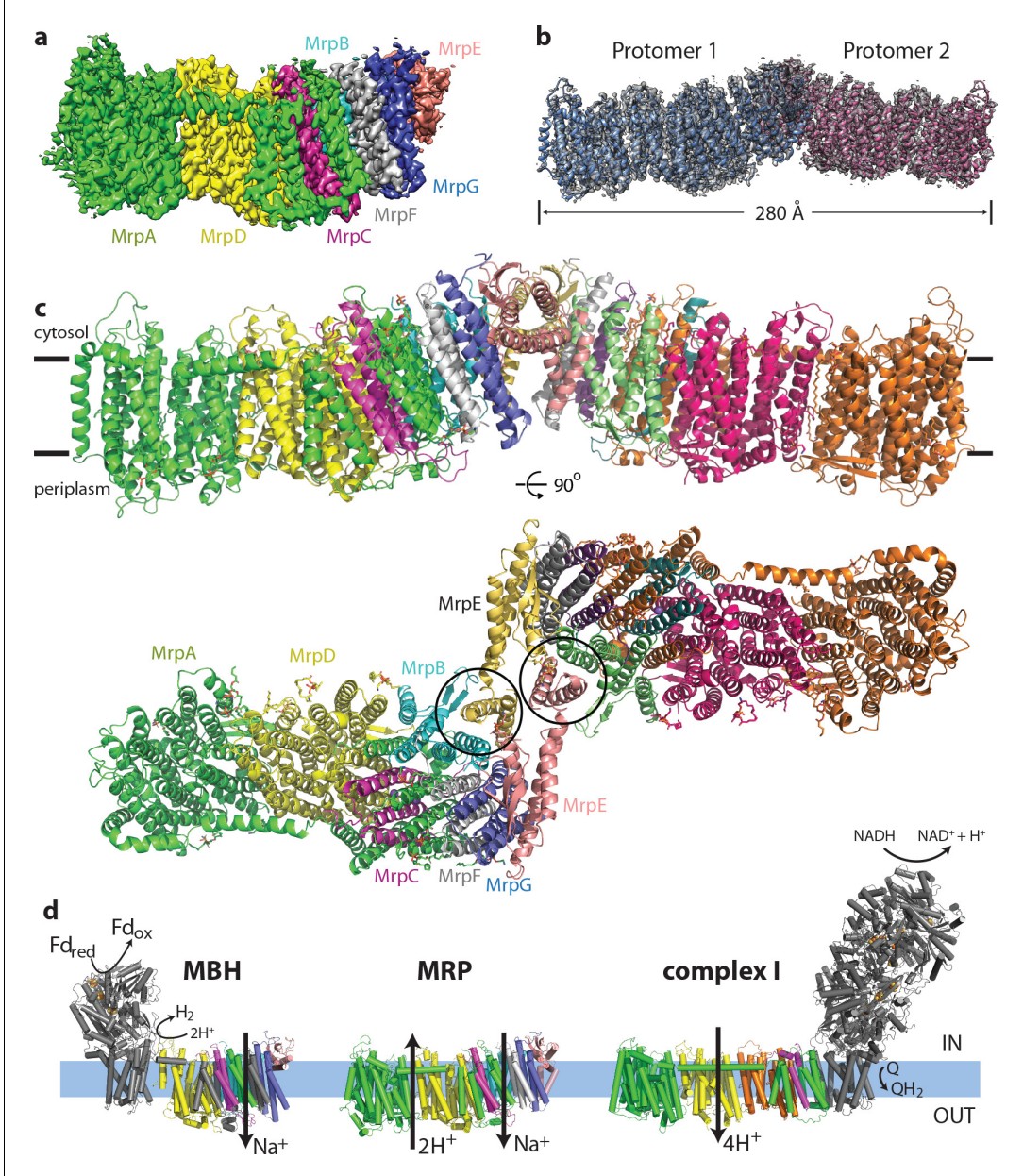

**Figure 1.** Overall structure of the Mrp complex. (**a**) Cryo-EM density of a monomer, coloured by subunit as indicated. (**b**) Cryo-EM density of a dimer, with the model shown as a cartoon. (**c**) Model of the dimer, with each subunit coloured differently. Side view and view from the cytoplasm, where two N-terminal helices of subunit MrpE, forming most of the dimer interface, are circled. (**d**) Schematic view of the Mrp monomer, MBH complex and complex I. Homologs of Mrp subunits are coloured similarly as in c), with an additional MrpD-like subunit in complex I in orange.

The online version of this article includes the following figure supplement(s) for figure 1:

**Figure supplement 1.** Purification and characterization of *Anoxybacillus flavithermus* Mrp.

**Figure supplement 2.** Processing of the Mrp-DDM dataset.

**Figure supplement 3.** Processing of the Mrp-LMNG dataset.

**Figure supplement 4.** Examples of cryo-EM density of Mrp monomer in LMNG.

protein purified in n-Dodecyl β-D-maltoside (DDM) detergent. Particles appeared as dimers of only approximate C2 symmetry as the angle between monomers varied, resulting in several 3D classes differing by that angle. After symmetry expansion in C2 point group in Relion, resulting pseudo-monomer particles could be refined to 3.4 Å resolution (*Figure 1—figure supplement 2*). This allowed initial model building for most of the model, however, cryo-EM density at the edges of the

monomer was fuzzy, with some TM helices (TMH) in the distal part of MrpA completely disordered. Therefore we purified the complex in a milder detergent Lauryl Maltose Neopentyl Glycol (LMNG) and collected a dataset again at 35˚ tilt. In this case dimers were overall 'flatter' than in DDM and possibly closer to their native shape in the flat lipid bilayer (*Figure 1—figure supplement 3*). Comparisons of cryo-EM maps of various dimers did not reveal any specific differences in the overall structure, apart from different apparent angles between the monomers. The best dimer class refined to 3.7 Å resolution (*Figure 1b*), while the best pseudo-monomer (after symmetry expansion) class refined to 3.0 Å with excellent density in all areas including previously disordered edges (*Figure 1a*). This allowed us to build and refine a high quality atomic model of the Mrp dimer (97% complete with only few terminal residues missing, *Table 1*, *Supplementary file 1*, *Figure 1c*).

## Overall structure

Each monomer consists of seven subunits with a total MW of 213 kDa and comprising 50 TM helices. The largest two subunits, including the N-terminal part of MrpA (MrpA$^N$, TMH1-16) and MrpD are arranged next to each other in a very similar way as the APLS of complex I (*Efremov and Sazanov, 2011*), with the small subunits BCEFG and the C-terminal part of MrpA (MrpA$^C$, TMH17-21) homologous to MBH subunits ABCDFG (*Yu et al., 2018*), attached 'on the right' of MrpD as shown in *Figure 1*. MBH subunit H is homologous to MrpD and MBH has its redox module attached 'on the left' of MbhH/MrpD, in contrast to complex I with the redox module on the right (*Figure 1d*). Thus, Mrp has nearly all subunits, except MrpA$^N$, in common with MBH. The fold of these Mrp subunits is extremely well preserved in MBH, including all the key residues, which are conserved and essential for activity (*Figure 3a,c*, *Supplementary file 1*). Similarly to MBH, subunits MrpC, A$^C$, F and G each fold into a three-helix sheet-like structure, forming four contiguous layers, flanked on one side by a four-helix subunit MrpB (*Figures 1c* and *2d*). MrpE caps the structure and is involved in dimerization.

The similarity to complex I extends to the entire MrpA, MrpD and MrpC subunits, with the fold and many key residues well conserved (*Figure 3a,b*, *Supplementary file 1* and *Figure 4—figure supplement 2b*). MrpA braces the complex with the long amphipathic helix HL (*Figure 2d*) extending from TMH15 to 16, similar to the arrangement in complex I. Among the three copies of APLS in complex I, MrpA is more homologous to the complex I subunit Nqo12/NuoL/ND5 (*T. thermophilus/E. coli*/mitochondrial nomenclature), while MrpD is closer to complex I Nqo14/NuoN/ND2 (*Figure 4—figure supplement 2b*). Similarly to APLS in complex I (*Efremov and Sazanov, 2011*), MrpA and MrpD contain N-terminal (TMH4-8) and C-terminal (TMH9-13) proton translocation half-channels that are related to each other by pseudosymmetry. Both half-channels contain lysine residues sitting on symmetry-related discontinuous (interrupted by a ~6 residue loop) helices (LysTMH7 and LysTMH12). These residues are likely the key to proton translocation because they are absolutely conserved, essential for activity and sit in a strategic position in the centre of each half-channel cavity (*Sazanov, 2015*). Key LysTMH7 forms a pair with a conserved TMH5 glutamate (GluTMH5), which is thought to modulate the pK$_a$ of lysine (*Efremov and Sazanov, 2011*). A central key lysine sits on another broken (by a π-bulge) helix TMH8 and connects the half-channels (*Figure 4a,b*). As in complex I, these key residues are connected by additional polar residues and form a central hydrophilic axis running through the middle of the membrane across the entire complex (*Figure 4a,b*). The flexibility provided by the broken helices may have a role in the conformational coupling mechanism and/or could help with pKa modulations of key residues.

As expected from sequence homology, the C-terminal MrpA TMH17-21 have the same fold as the complex I subunit Nqo10/NuoJ/ND6, although the three essential for activity carboxylate residues from MrpA, E706, D792 and E796 (*Supplementary file 1*, *Figure 4—figure supplement 3*), are not conserved in complex I (*Figure 4—figure supplement 2b*). Key E706 is replaced by Y59 in complex I, where this essential tyrosine sits on a π-bulge in TMH3 of Nqo10. This π-bulge can be wound up in mammalian complex I by a striking rotation of this helix, highly conserved in complex I and probably critical for its mechanism (*Agip et al., 2018*; *Letts et al., 2019*). The π-bulge in the corresponding helix is also present in the MBH complex (*Yu et al., 2018*). Interestingly, the bulge appears to be absent in the Mrp, so it may not be a universal feature or it is possible that it may appear in a different state of the complex, for example at high pH (current structure is solved at pH 6.0). Subunit MrpC fold is very similar to the complex I subunit Nqo11/NuoK/ND4L but instead of

**Table 1.** Cryo-EM data collection, refinement and validation statistics.

**Data collection and processing**

| | | |
|---|---|---|
| Magnification | 105000 | |
| Voltage (kV) | 300 | |
| Electron exposure (e⁻/Å²) | 85 | |
| Defocus range (μm) | 0.6–2.3 | |
| Pixel size (Å) | 0.84 | |
| Symmetry imposed | C1 | |
| Initial particle images (no.) | 889272 | |
| Final particle images (no.) | 285688 | |
| Map resolution (Å) | 2.98 | |
| FSC threshold | 0.143 | |
| Map resolution range (Å) | 2.85–4.2 | |
| **Refinement** | **Monomer** | **Dimer** |
| Initial model used (PDB code) | 6CFW, 4HEA | 6CFW, 4HEA |
| Model resolution (Å) | 2.97 | 3.04 |
| FSC threshold | 0.5 | 0.5 |
| Map sharpening B factor (Å) | −42, local resolution-filtered | −42 |
| Model composition | | |
| Non-hydrogen atoms | 15448 | 30250 |
| Protein residues | 1873 | 3730 |
| Ligands | 510 | 1020 |
| Waters | 242 | 0 |
| B factors (Å²) | | |
| Protein | 88.4 | 74.4 |
| Ligand | 139.1 | 123.4 |
| Waters | 89.8 | N/A |
| R.m.s. deviations | | |
| Bond lengths (Å) | 0.0080 | 0.0078 |
| Bond angles (°) | 1.32 | 1.33 |
| Validation | | |
| MolProbity score | 1.47 | 1.31 |
| Clashscore | 3.58 | 2.17 |
| Poor rotamers (%) | 0.13 | 0.10 |
| Ramachandran plot | | |
| Favoured (%) | 95.44 | 95.45 |
| Allowed (%) | 4.56 | 4.55 |
| Disallowed (%) | 0 | 0 |
| EMRinger score | 3.73 | 3.61 |

two histidines H37 and H40, conserved in Mrp, complex I has two essential glutamates (*Figure 4— figure supplement 2b*).

A striking feature of the Mrp complex is the tightly intertwined interface between the two monomers. To allow this, the two N-terminal TMH of MrpE (circled in *Figure 1c*) have swung out from their respective positions in MBH (*Figure 3a* top), resulting in extensive interactions between subunits MrpE and MrpB from both monomers. This shows that the Mrp complex clearly evolved to exist

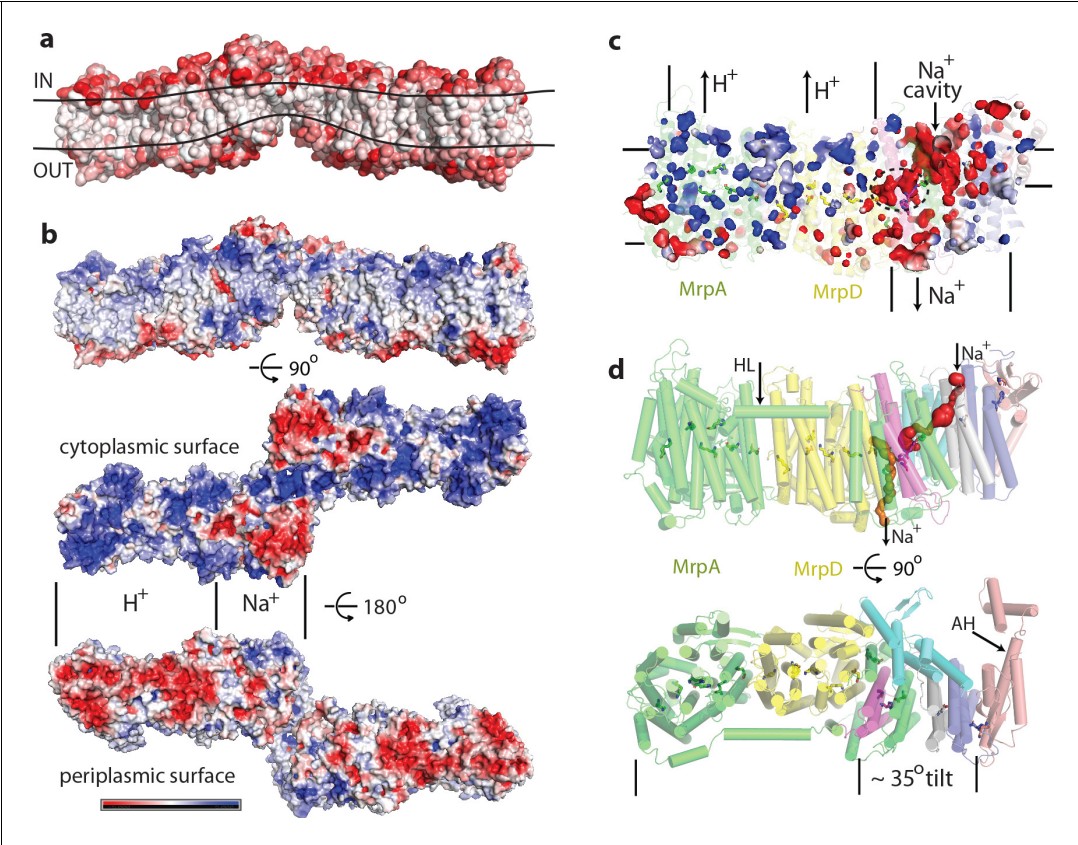

**Figure 2.** Physicochemical properties of the Mrp complex. (**a**) Surface of the dimer is shown with residues coloured according to Eisenberg hydrophobicity scale from white (hydrophobic) to red (hydrophilic). Apparent extent of the lipid membrane is outlined. (**b**) Surface charge distribution of the dimer. Top is side view with cytosolic side up. Protein surface is shown coloured red for negative, white for neutral and blue for positive surface charges, with the scale (-5 to +5 $k_bT/e$) shown below. Likely areas of interactions with protons and sodium and are indicated. (**c**) Internal cavities in the monomer, coloured according to charge. (**d**) Top, side view of the monomer with tunnels calculated in MOLE shown as red and orange surfaces. Amphipathic helix HL from subunit MrpA is indicated. Bottom, view from the cytoplasm illustrating the high degree of tilt of all helices in the putative $Na^+$-translocating domain relative to helices in the $H^+$-translocating domain ($MrpA^N/D$). Amphipathic helix AH from subunit MrpE is indicated. The online version of this article includes the following figure supplement(s) for figure 2:

**Figure supplement 1.** Local resolution, FSC curves, bound ions and membrane thinning.

as a dimer in vivo, as also confirmed by the stability of the dimer contacts in all the 3D classes that we observe and the dimer stability during purification in several species (Materials and methods and *Morino et al., 2014*). Another notable feature of the structure is a dramatic tilt of about 35° of all the TM helices in the small subunits, as compared to $MrpA^N$ and MrpD (*Figure 2d*). Such an unusual fold is probably stabilised by the dimer architecture since the helices are tilted in opposite directions at the interface where two protomers meet (*Figure 1c*).

Another striking feature is a very unusual surface distribution of hydrophobic lipid-exposed residues – starting at the edges of the dimer from the standard ~30 Å wide hydrophobic belt as needed to reside in the membrane, at the interface between the monomers the apparent belt thins to about 15–20 Å. This is obvious from both the hydrophobicity of surface residues and from the calculated surface charge distribution (*Figure 2a,b*). Consistently, the detergent belt, visible in low-resolution maps (*Figure 2—figure supplement 1h*), gets very thin in this area. Overall the exposed hydrophobic belt is roughly linear (*Figure 2a*), consistent with the flatness of lipid bilayer. This suggests that these dimers, with about 20 degrees apparent angle between the monomers, which refined to the highest resolution (*Figure 1—figure supplement 3*), are probably close to the physiological state of the complex.

The thinness of the dimer interface is mainly due to the two very short N-terminal TMHs of MrpE, of only about 15 residues each, which form the bulk of the interface, resulting in only ~20 Å total

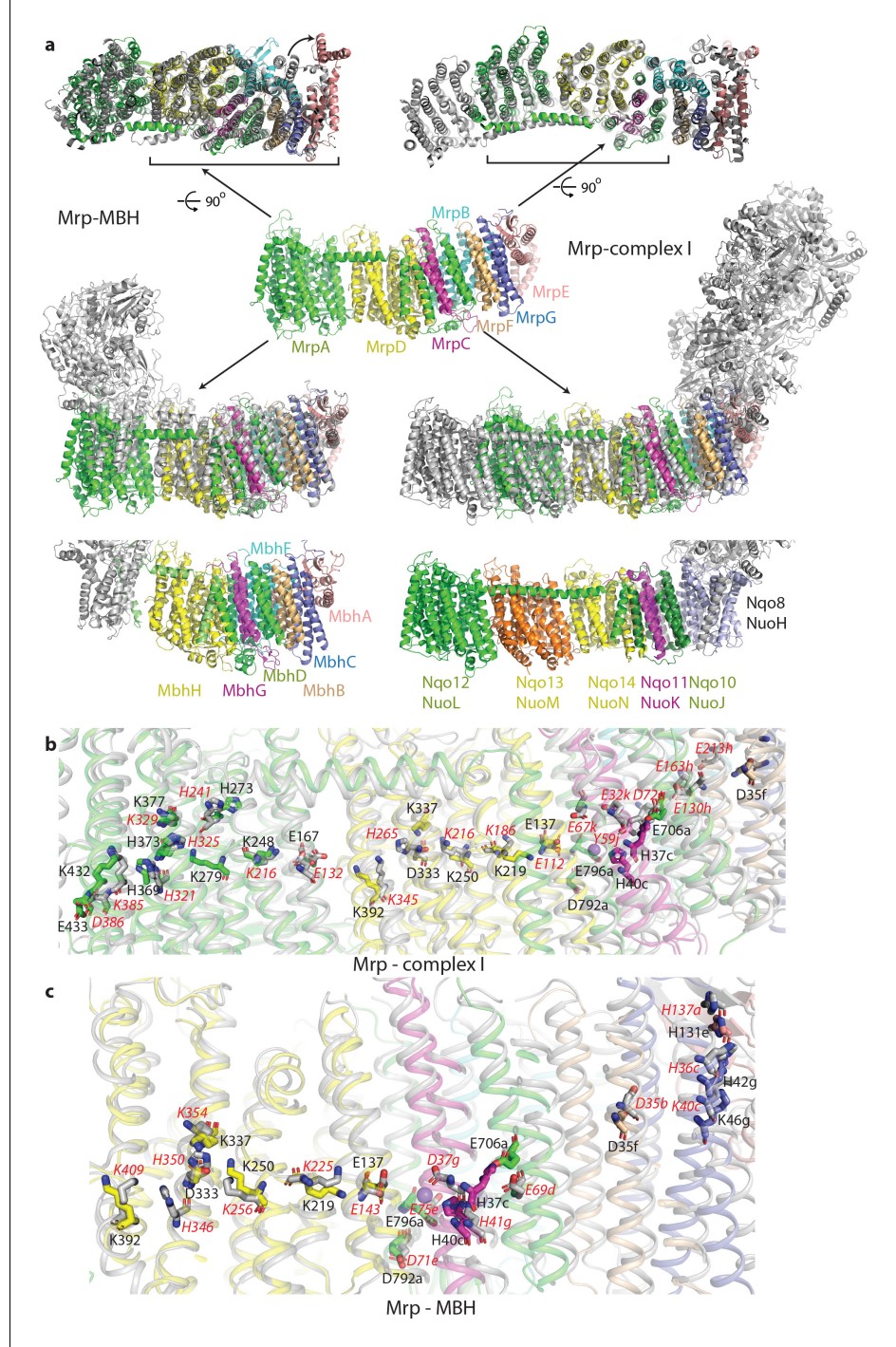

**Figure 3.** Overlay of Mrp with complex I and MBH complex. (**a**) The Mrp monomer is shown in the centre in side view. Alignments to MBH are in the left column and to complex I in the right column. The top row shows view from the cytosol, with the conserved domains underlined. The arrow indicates 'swinging out' of the MrpE TMH1-2 in comparison to MBH. The second row shows the side view, with Mrp subunits coloured as in the centre and complex I/MBH in grey. In the bottom row only MBH and complex I are shown in the same orientation as above, with subunits homologous to Mrp coloured as in Mrp and the rest grey. Quinone-binding subunit in complex I (Nqo8/NuoH) is not present in Mrp and is highlighted in slate. Additional MrpD-like subunit in complex I (Nqo13/NuoM) is in orange. MBH has its redox module attached to the Mrp-like domain on the opposite side compared to complex I. MbhF is homologous to MrpB, MbhG to MrpC, MbhH to MrpD, MbhA to MrpE, MbhB to MrpF, MbhC to MrpG and MbhD together with MbhE to MrpA$^C$. Nqo12/NuoL is homologous to MrpA$^N$, Nqo13/NuoM

*Figure 3 continued on next page*

*Figure 3 continued*

and Nqo14/NuoN to MrpD, Nqo11/NuoK to MrpC and Nqo10/NuoJ to MrpA$^C$. (b) Overlay of Mrp with the membrane domain of complex I. Key residues, important for H$^+$ translocation and Na$^+$ binding are shown in stick representation, with the residue number indicated in black for Mrp and the corresponding homologous residue of complex I indicated in red. Except for MrpA$^N$ and MrpD, suffixes indicate subunit (in *E. coli* nomenclature for complex I). (c) Overlay of Mrp with the homologous subunits of MBH. Key residues, important for H$^+$ translocation and Na$^+$ binding are shown in stick representation, with the residue number indicated in black for Mrp and the corresponding homologous residue of MBH shown in red. Except for MrpD, suffixes indicate subunit. The positions of two bound K$^+$ ions in Mrp are indicated by purple spheres, with one of them being less visible behind H37*c*.

protein span across the membrane. These helices are continued in MrpE by a long amphipathic helix (AH, *Figure 2d*), which likely resides at the surface of the membrane (*Figure 2—figure supplement 1g*). This helix is continued by a ferredoxin-fold domain of MrpE, which is exposed to the cytoplasm along with parts of TM helices of MrpG. A very thin hydrophobic belt is clearly unfavourable for membrane protein folding and so it must have a functional role in Mrp, most likely by acting to significantly disturb and thin a lipid membrane in this area. Such a thinning could be important for the shortening of the path along which Na$^+$ ions have to be moved across the membrane against the electric potential. The thinning of the membrane could be achieved by a combined action of the short MrpE TMH1-2 from both monomers (*Figure 1c*) driving the long amphipathic helices of MrpE into the bilayer. The highly tilted helices of the small subunits could also help to achieve and sustain the thinning as they approach the dimer interface.

The surface charge distribution on the hydrophilic surface of Mrp is also very striking. On the cytoplasmic surface, the MrpA$^N$/MrpD area is uniformly positively charged, while that of the small subunits is negatively charged (*Figure 2b*). This pattern is completely reversed on the periplasmic side. This is very appropriate for the MrpA$^N$/D subunits, which are likely to be involved in proton translocation, as negative charge in the periplasm will help the protein to gather protons that are scarce at high pH. On the other hand, positive charge on the inside surface of MrpA$^N$/D will help to release translocated protons into the cytoplasm. The negatively charged area over the small subunits would be suitable for attraction of Na$^+$ or K$^+$ ions from the cytoplasm, while the positive charge on their periplasmic side is suitable for the release of translocated ions. This pattern is striking also for internal cavities – the MrpA$^N$/D area has many small positively charged voids, while the area of the small subunits is full of negatively charged cavities, with some of them being quite large (*Figure 2c*).

This strikingly bipartite pattern of charge distribution strongly suggests that MrpA$^N$/D subunits are involved in proton translocation into the cell, while the entire domain encompassing MrpBCEFG and MrpA$^C$ is involved in sodium translocation in the opposite direction (as indicated in *Figure 2b, c*). We discussed previously that sodium might be translocated at the interface between MrpA$^N$ and MrpD (*Sazanov, 2015*). However, the structure reveals that this interface is tightly sealed, hydrophobic and does not contain any suitable cavities, excluding it as a candidate for a Na$^+$-binding site. Analysis of potential cavities and channels by the MOLEonline software (*Pravda et al., 2018*) revealed that most of the potential tunnels originate in two large neighbouring cavities found near the conserved H37*c* and H40*c* (we will indicate with italic suffix the subunit, in this case MrpC) (*Figure 4—figure supplement 1f*, *Figure 2c,d*). An additional cavity is found between MrpA$^C$ and MrpF subunits. This is also the largest cavity in Mrp and contains several bound lipids (modelled as phosphatidylethanolamine, *Figure 4—figure supplement 1a*) but the lipids do not fill the cavity entirely, leaving plenty of space. The cavity is present in both LMNG and DDM sample maps and in all types of dimers with different angles between monomers, confirming that it is not an artefact of a particular solubilisation condition or conformational state. We suggest that since the cavity is highly negatively charged (*Figure 2c*) and lined with some polar residues (including Y684*a*, T707*a* and T72*f*), it could be used for passage and temporary storage of translocated cations (we will call it Na$^+$ cavity and it is indicated as such in *Figure 2c* and as *NaC* in *Figure 4*).

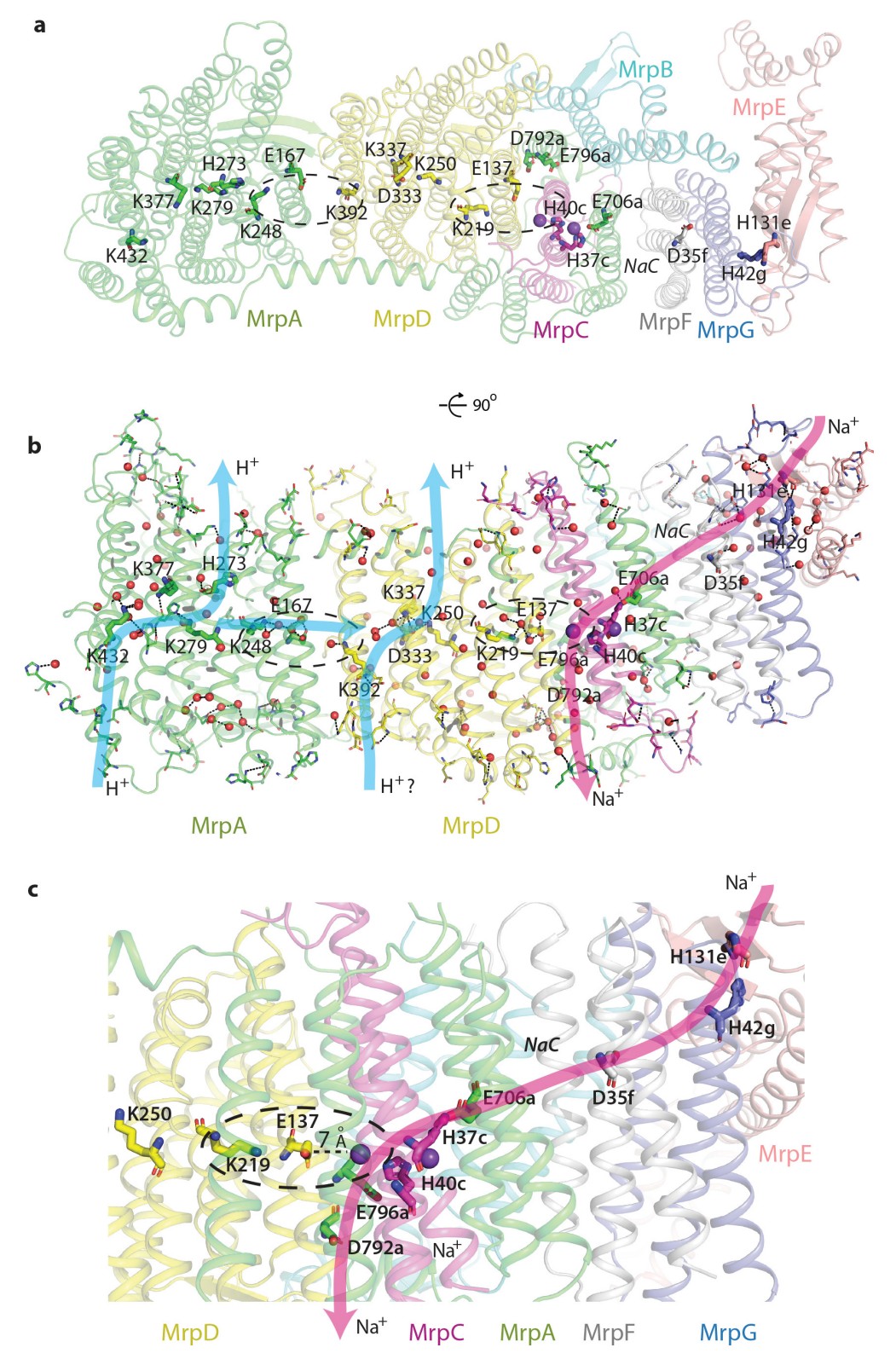

**Figure 4.** Proton and cation translocation pathways. (a, b) View from the cytoplasm (a) and side view (b) with key residues proposed to be involved in both pathways indicated. Waters predicted in Dowser software are shown as red spheres and the hydrogen bonds involving protonatable residues and waters are shown as black dashes. Approximate pathways for $H^+$ and $Na^+$ translocation are indicated by arrows. Coupling points between MrpA[N] and MrpD, and between MrpD and $Na^+$-binding site are indicated by dashed ovals. (c) Details of $Na^+$ pathway. The two experimentally identified bound $K^+$

*Figure 4 continued on next page*

*Figure 4 continued*

ions are shown as violet spheres. The key distance between one of the ions and GluTMH5$d$ in the Na$^+$/H$^+$ coupling point is indicated. A large cavity between MrpA$^C$ and MrpF is indicated as *NaC*.

The online version of this article includes the following figure supplement(s) for figure 4:

**Figure supplement 1.** Bound lipids, waters and ions; Na$^+$ pathway.
**Figure supplement 2.** Conservation patterns.
**Figure supplement 3.** Sequence alignment of Mrp subunits.
**Figure supplement 4.** Sequence alignment of Mrp subunits.

One of the prominent tunnels identified by MOLE (red in *Figure 2d*) originates near H37$c$, passes by the key E706$a$, through the Na$^+$ cavity and then near the conserved D35$f$ before exiting into the cytoplasm. It is negatively charged at the origin, as would be appropriate for Na$^+$, but has some narrow passages of about 1 Å radius (*Figure 4—figure supplement 1d*), so the access must be regulated. Another tunnel (orange in *Figure 2d*) originates in the cavity near H40$c$, which is also lined by the key E137$d$ (GluTMH5) and continues towards the periplasm at the interface between MrpD and MrpC subunits (*Figure 4—figure supplement 1e*). Since H37$c$ and H40$c$ cavities are linked, these two tunnels can form a channel for Na$^+$ translocation across the membrane, with the passage through the narrow restrictions regulated as part of the coupling mechanism.

## Proton translocation pathways

A striking pattern of charge distribution and conservation of the fold and key residues between proton-pumping complex I and Mrp (*Figure 4—figure supplement 2b*) leaves no doubt that proton translocation channels are found in MrpA$^N$ and MrpD subunits, similarly to complex I and other related redox proton pumps. High local resolution (~2.9 Å) in the core of the structure allowed us to model many bound water molecules (*Figure 2—figure supplement 1a,f*). They are found mostly on the hydrophilic surfaces of the complex and along the entire central hydrophilic axis around key charged residues, confirming the previously suggested hydration of the central axis (*Baradaran et al., 2013*; *Figure 4—figure supplement 1b*). Since at 2.9 Å resolution we are still limited in the identification of water molecules, for the analysis of the complete proton translocation pathways we have modelled waters in Dowser (*Zhang and Hermans, 1996*; *Figure 4b*). Many experimental and Dowser-predicted waters coincided, but some were identified only by Dowser. The analysis of connections between Grotthus-competent residues (K, H, E, D, T, S and Y [*Khaniya et al., 2020*]) and waters (allowing for Grotthuss mechanism of proton transfer) revealed that the highly hydrated cluster near MrpA LysTMH12 (K432) and LysTMH8 (K279), containing also conserved H273, H369, and K377 (*Figures 3b* and *4b*), is all interconnected and linked to the periplasm via the conserved E433. The link to the cytoplasm is most likely along H273, which sits on TMH8. The potential link is not continuous and must exist only temporarily during the catalytic cycle, allowing for protons to be ejected into the cytoplasm. Analyses of MrpA$^N$ and MrpD and comparisons with known complex I structures suggest that in APLS, links to the cytoplasm are achieved not along the centre of the N-terminal half-channel as discussed originally (*Efremov and Sazanov, 2011*), but mostly along TMH8, which has more polar residues in the area. TMH8 is suited to play a functional role due to its π-bulge which is conserved in all APLS. The key LysTMH8 interacts with the backbone oxygen of the π-bulge and is the only protonatable residue on TMH8 in MrpD and MrpD-like subunits of complex I (Nqo13/NuoM/ND4 and Nqo14/NuoN/ND2), while in MrpA H273 is added to the π-bulge and only this histidine is conserved in Nqo12/NuoL/ND5, presumably replacing LysTMH8.

The area between LysTMH12 (K392) and LysTMH8 (K250) in MrpD is also highly hydrated and interlinked with participation of D333 and K337, which are replaced by one histidine in complex I Nqo13 and Nqo14 subunits (*Figure 3b*). However, in contrast to MrpA$^N$, LysTMH12$d$ does not seem to be linked to the periplasm due to the lack of polar residues and waters (either experimental or Dowser-modelled) in this area. An analogous situation appears to exist in complex I, where only the MrpA-like subunit Nqo12/NuoL/ND5 is clearly linked to the periplasm in nearly identical to MrpA arrangement (*Figure 3b*), while Nqo13 and Nqo14 apparently lack such links, as similar analysis shows. Therefore, it is possible that from the periplasm side all the protons enter the Mrp complex

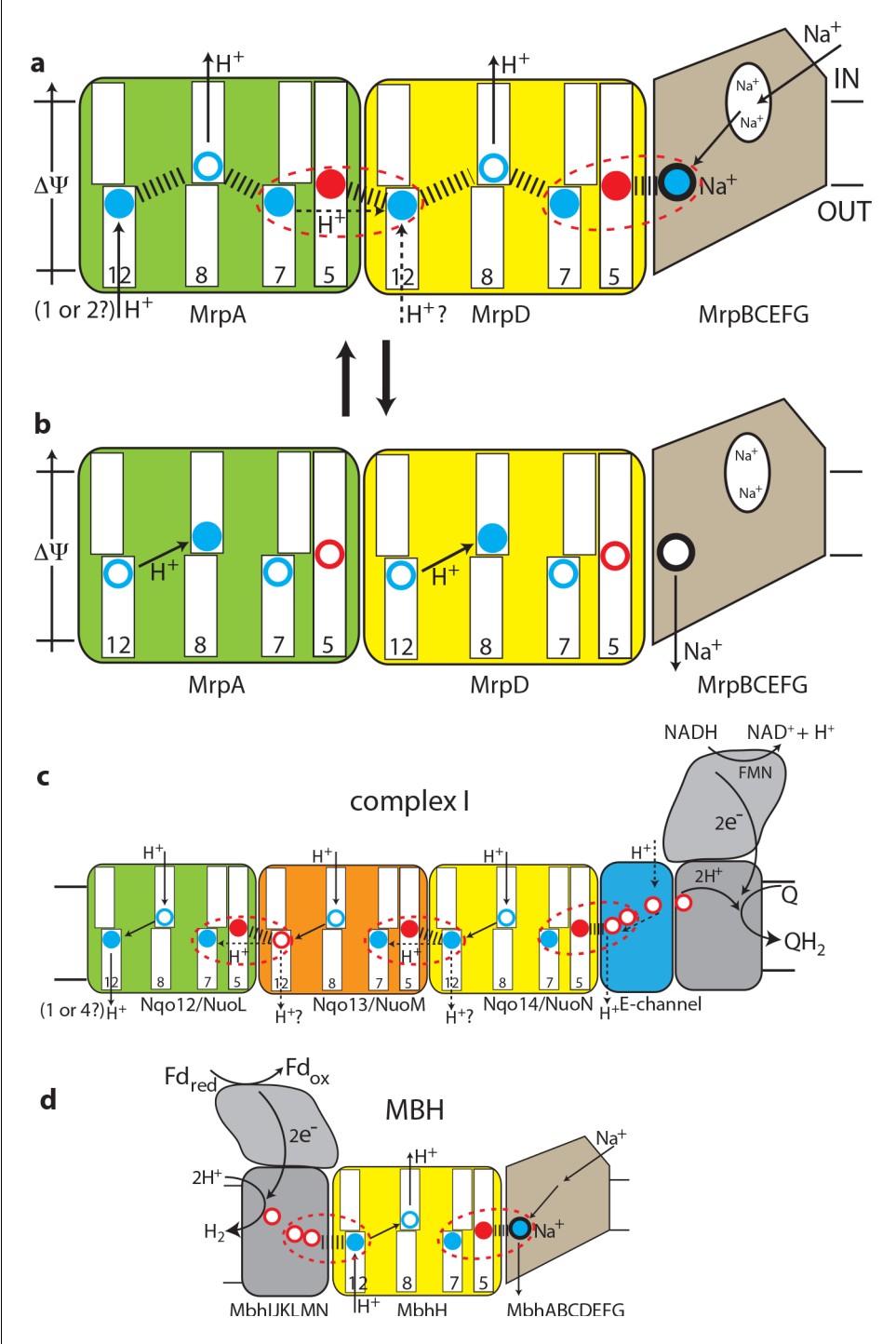

**Figure 5.** The antiport mechanism of the Mrp complex and its conservation in complex I and MBH. (**a**) A likely state of the Mrp complex in the current structure, as a part of the catalytic cycle. Key glutamates from the central axis are shown as red circles and key lysines as blue circles, with charged state depicted by filled circle and neutral state by empty circle. Key helices where these residues sit are numbered and shown as broken where applicable. Coupling points are indicated by red dashed ovals. Electrostatic interactions are indicated as thick dashed lines. Possible lateral re-distribution of protons between MrpA and MrpD is indicated as alternative to the direct entry of protons to MrpD from the periplasm. Na$^+$ ion (blue) is bound in the coupling point and the large Na$^+$ cavity mid-pathway is indicated as an empty oval. A sequence of events leading up to this state is described in the text. (**b**) An alternative state of the Mrp complex as another part of the catalytic cycle. Na$^+$ ion is expelled into the periplasm and coupling points become neutral. (**c**) Complex I contains one additional MrpD-like subunit (orange, Nqo13/NuoM), with the change of LysTMH12 to glutamate. The coupling points between the APLS are conserved and the coupling point with Na$^+$ is replaced by coupling to the charge of glutamates in the

*Figure 5 continued on next page*

*Figure 5 continued*

E-channel. (**d**) MBH instead of a MrpA-like subunit contains a [NiFe] hydrogenase module, which oxidises ferredoxin and reduces protons to hydrogen. This module has a chain of glutamates similarly to complex I and their charge likely boosts proton translocation into the cell (opposite direction to complex I) via the modified coupling point (red dashed oval on the left). The $H^+/Na^+$ coupling point and mechanism are retained from the Mrp.

via MrpA$^N$ subunit (and leave complex I via Nqo12 subunit). In contrast, links to the cytoplasm along TMH8 could be functional in all APLS, although they are much more populated by polar residues in MrpA$^N$ and Nqo12 or Nqo13 compared to MrpD and Nqo14. There is a distinct possibility therefore that protons entering via MrpA$^N$ are re-distributed to MrpD via the abundantly hydrated central axis, as indicated in *Figure 4b*, and entry from the periplasm via MrpD is questionable, as also indicated. In complex I, which normally works in the opposite direction, protons from the cytoplasm may be entering mainly via Nqo12 and Nqo13, get redistributed along the central axis and then pumped into the periplasm mainly via Nqo12. Although these new MrpA$^N$-only-H$^+$-in or Nqo12-only-H$^+$-out models now appear to be more likely than the traditional view that each antiporter-like subunit carries one proton fully across the membrane (*Efremov and Sazanov, 2011*), the distinction between the two models remains to be established in future experiments. It is, however, not important for the mechanism that we propose below, since the re-distribution of protons along the highly hydrated central axis is likely to be fast and not rate-limiting.

In this scenario the key LysTMH7/GluTMH5 pairs participate in the re-distribution of protons along the central axis but do not seem to play a large role in proton pathways, as those are formed from residues around LysTMH12 and LysTMH8. What is the role of these pairs then? The residues in Lys/Glu pairs are close enough to each other for strong electrostatic interactions but not close enough to form salt bridges, as they are separated by 6–7 Å. Since they are invariant and essential we propose that their main role is to regulate and control proton pathways via electrostatic interactions. The entire complex appears to be finely tuned electrostatically (*Figure 2b,c*) and the pair is close enough to LysTMH8 (~10–15 Å) for changes in the LysTMH7 charge state to be able to change the p$K_a$ of LysTMH8. Additionally, MrpA GluTMH5 is only 7 Å away from MrpD LysTMH12 and so these two residues would influence each other's p$K_a$. Therefore, LysTMH7 and GluTMH5 from one APLS together with LysTMH12 from the neighbouring APLS probably represent coupling points in the mechanism, both in Mrp (circled in *Figure 4a,b*) and in complex I. Charge switches in these points would control proton access to the periplasm via LysTMH12 (either directly or indirectly via proton re-distribution) and to the cytoplasm via electrostatic interactions with LysTMH8. A similar role for Lys/Glu pairs was suggested previously on the basis of MD simulations (*Di Luca et al., 2017*). However, in the MD study the change in distance between lysine and glutamate was considered as a key 'switch', while we propose as a driving force a more robust, in our opinion, change in their protonation state.

## Cation (Na$^+$ or K$^+$) translocation pathway

Our cryo-EM maps revealed strong cation density (stronger than waters) both in DDM and LMNG samples (Materials and methods, *Figure 2—figure supplement 1e,f*). The cations are coordinated by the conserved H37$c$ and H40$c$ within the two large hydrophilic cavities (which we will call double cavity) surrounding these residues (*Figure 4—figure supplement 1c,f*). Histidines can coordinate both Na$^+$ and K$^+$ (*Zheng et al., 2008*) and we assigned the cations as K$^+$ ions since we had K$^+$ in the buffer and the coordination pattern is consistent with K$^+$ (Materials and methods). It can be expected that Na$^+$ ions will bind in the same positions, with the coordination completed by the conserved S36$c$, S80$c$, N702$a$ and several waters (*Figure 4—figure supplement 1c*). One of the bound ions can interact electrostatically with key E137$d$ (GluTMH5) and another one with key E706$a$ (*Figure 4—figure supplement 1c*), so they are in a really strategic position for coupling proton and cation translocation. Site-directed mutagenesis performed on Mrp complexes confirmed the essential role of the key APLS residues both for proton and cation translocation (*Supplementary file 1*). The essential role of MrpA$^C$ E706, D792 and E796 was confirmed in several species. Mutations in H719, coordinating the headgroup of the lipid bound in the Na$^+$ cavity, in P721, sitting at the end of the helix containing E706, in key E137$d$, the nearby F136$d$, and in P114$e$ specifically affected $K_m$ for

Na$^+$, indicating a role for these residues in the Na$^+$ pathway. P114e sits on a loop in the ferredoxin-like domain of MrpE, which also contains H131e near H32g. These conserved histidines may bind Na$^+$ in the part of MrpE/G near the AH helix. This area at the tip of the monomer, where the membrane is likely to be thinned, is as highly conserved as the core of APLS, attesting to its functional importance (*Figure 4—figure supplement 2a*). It is also the most negatively charged area on the cytoplasmic surface (*Figure 2b*) and is likely to represent the entry point for cations. Conserved D29f is one of the residues responsible for the negative charge here and its mutation to alanine completely abolished the activity (*Morino et al., 2010*).

On the basis of these considerations and according to the analysis of cavities and tunnels by MOLE as described above, we propose that the Na$^+$ pathway (which applies also to K$^+$) starts near H42g/H131e, passes by the highly conserved D35f, traverses the large Na$^+$ cavity between MrpF and MrpA$^C$, passes by the key E706a and enters the double cavity where at least two cations, as we observe, can be coordinated between E706a, H37c, H40c and E137d (*Figure 4c*, *Figure 4—figure supplement 1c,f*). The double cavity most likely represents a point of coupling between proton and cation translocation (circled in *Figure 4c*), as here Na$^+$ ion(s) sit in a position analogous to the LysTMH12 in APLS coupling points. This way, the proton translocation through MrpD (already coupled to MrpA$^N$ via K248a/E167a/K392d coupling point) can directly be coupled to Na$^+$ translocation. The Na$^+$ path continues towards the periplasm at the interface between TMH5d, TMH3c and TMH21a, ending with the key E796a and D792a at the exit into the periplasm (*Figure 4c*). T75c is facing E796a and is essential for growth in high salt (*Supplementary file 1*).

The role of conserved D35f was not clear from previous mutations in *Bacillus subtilis* Mrp, as substitutions were highly detrimental when the complex was expressed in *E. coli* (perhaps the assembly of the complex was affected) but not so in *B. subtilis* itself (*Kajiyama et al., 2009*). However, in *B. subtilis* only conservative mutations to polar residues E and N were explored and so the effects could be expected to be mild. To clarify this, we generated mutants of *A. flavithermus* Mrp complex by mutating D35f to L, a hydrophobic residue of roughly similar size to D, and expressed the complex in *E. coli* similarly to WT Mrp. The mutation completely deactivated the complex as shown by the absence of growth at high NaCl concentrations (*Figure 1—figure supplement 1j*). The dimer seemed to be de-stabilised as well, although the monomer was fully assembled (*Figure 1—figure supplement 1d,e*). To try to clarify how important the dimer is for the function, we generated MrpE L41W and MrpG S72W mutants. They were chosen because there are no salt bridges linking the monomers but these residues form close contacts between the two monomers and we expected that an introduction of bulky tryptophanes may disrupt the dimer. This, however, did not happen and the activity was not affected (*Figure 1—figure supplement 1d,e,j*). Therefore, the dimer can withstand some perturbations but it may not be absolutely essential for the mechanism per se, as a very similar fold exists as a monomer in MBH. The role of the Mrp dimer may have more to do with the stabilisation of the unfavourable membrane-thinning fold as discussed above. Nevertheless, our data suggests that invariably conserved D35f is important for activity, consistent with its position on the Na$^+$ path (*Figure 4c*). Furthermore, this aspartate is found right opposite the invariable and essential for activity (*Morino et al., 2010*) P86g, which breaks TMH3 of MrpG in half and bends it at a point where this helix contacts the MrpE ferredoxin-fold domain with its essential P114. Such architecture suggests that conformational interactions are important in this area, perhaps as a part of gating for Na$^+$ entry and exit.

Summing up, surface and cavity charge distribution, analysis of channels and cavities, patterns of sequence conservation and mutagenesis results, cryo-EM density for bound cations and basic mechanistic considerations all overwhelmingly support the Na$^+$ path as depicted in *Figure 4b,c*. The key residues are arranged as a kind of ladder descending from the cytoplasm to the periplasm along the repeating pattern of the three-helix fold of the small subunits. In addition to the already discussed key residues, many further, mostly conserved, polar residues line the path all along the way, as expected for a cation pathway (*Figure 4—figure supplement 1f*). The path differs from the proposal for the MBH complex, where the suggested entry point roughly coincided with our view (around D35f), while the exit was proposed to be directly 'below', near D59f (D59b in MBH) (*Yu et al., 2018*). However, the area around D59f (not universally conserved residue) is exposed to the periplasm and is separated from D35f by several layers of highly hydrophobic residues, therefore this proposal for the Na$^+$ path is extremely unlikely. Instead, we propose that the Na$^+$ pathway in MBH is the same as in Mrp, as all the key residues and fold are very well conserved (*Figure 3c*). The

double cavity in the $Na^+/H^+$ coupling point is of similar size and also negatively charged in MBH. The only difference is that H40$c$ from Mrp is replaced by D37$g$ in MBH, a residue which is also capable of coordinating $Na^+$. D37$g$, H41$g$, E69$d$, N34$g$ and about six nearby serines are ideally arranged in MBH to coordinate two cations similarly to Mrp (*Figure 3c*). In fact, in the deposited cryo-EM density of the MBH complex (EMD-7468) there is a clear density for a potential $Na^+$ ion coordinated by E69$d$ (E706$a$ homologue). The cavity between MbhB (MrpF) and MbhD (MrpA$^C$) is also present but is not as large as the $Na^+$ cavity in Mrp, perhaps because of a lesser demand for temporary $Na^+$ storage in MBH. Residues that we propose to be involved in the coupling and exit points of the $Na^+$ path (from E706$a$ to D792$a$, *Figure 3c*) were instead suggested to be a part of an additional $H^+$ translocation path in MBH, working in the opposite direction to MbhH/MrpD (*Yu et al., 2018*). This is again very unlikely because the site in the coupling point is extremely well suited to be a $Na^+$ binding site due to all the overwhelming evidence listed above. The main argument in support for an additional $H^+$ translocation path in MBH came from the similarity to complex I, where this pathway (E-channel) is not well established and should be working in the same direction as MrpD/Nqo14 and not in the opposite. This is hardly strong evidence, along with the absence of any credible exit point into the cytoplasm, since the proposed direct path for $H^+$ from the double cavity towards the cytoplasm (*Yu et al., 2018*) is blocked by several layers of hydrophobic residues both in MBH and Mrp. In summary, we suggest that the $Na^+$ path depicted in *Figure 4* is common for Mrp and MBH and that the additional $H^+$ path proposed for MBH does not exist.

## Discussion

Importantly, a striking electrostatic imbalance of $H^+$ and $Na^+$ modules of Mrp (*Figure 2b,c*), the arrangement of key residues in the $H^+$ and $Na^+$ pathways, the separation of $H^+$ cross-pathways from the LysTMH7/GluTMH5 coupling points and a remarkable similarity of the MrpA$^N$/MrpD and MrpD/$Na^+$ coupling points (*Figure 4a,b*) all collectively suggest that electrostatic interactions are the main driving force in the antiport mechanism.

Overall however, the electrostatic forces are likely to be amplified by coordinated conformational changes, because most of the key residues (including all four LysTMH8 and LysTMH12) sit on breaks of TM helices and interact with exposed backbone oxygen atoms from such breaks, so even small movements of these helices will affect each other and $pK_a$'s of these residues. Therefore, the Mrp complex is likely to exist in two different conformational states, one of which is conductive to $Na^+$ binding from the cytoplasm and another is associated with $Na^+$ release into the periplasm. However, the difference between the states can be really minor, because changes in $pK_a$'s can be associated with only small changes in the local environment. In order to ascertain the charge state of the key residues in our current structure, we analysed PROPKA (*Bas et al., 2008*) predictions and the appearance of cryo-EM density, especially of E/D residues, since due to radiation damage carboxylates lose side-chain density in cryo-EM when they are in a charged state (but not if in a salt bridge) (*Baker and Rubinstein, 2010*; *Grant and Grigorieff, 2015*). These analyses suggest that both in MrpA$^N$ and MrpD LysTMH7/GluTMH5 pairs have both residues charged, LysTMH12 is charged and LysTMH8 is probably neutral. We also know that two cations ($K^+$ in our structure but usually $Na^+$ in vivo) are bound in the double cavity in the $H^+/Na^+$ coupling point, with one of them likely to be destined for translocation into the periplasm, meaning that there is one additional positive charge due to the bound cation (indicated as a filled blue circle and $Na^+$ in *Figure 5*).

These considerations allow us to arrive at a model for the Mrp state in the current structure as depicted in *Figure 5a* and a mechanism as follows. The distribution of charges in *Figure 5a* would result from $Na^+$ charge effectively lowering the pKa of MrpD GluTMH5 (7 Å away, *Figure 4—figure supplement 1c*), so that it donates its proton to MrpD LysTMH7 (6 Å away). At the same time both LysTMH12 would have been protonated from the periplasm. Positive charge on MrpD LysTMH12 lowers the pKa of MrpA GluTMH5 so that it donates its proton to MrpA LysTMH7, mirroring events in MrpD. Assuming LysTMH8 was protonated before, now the positive charge both on LysTMH7 and LysTMH12 will force it to lose the proton to the cytoplasm both in MrpA$^N$ and MrpD, giving us the state in *Figure 5a*. Next, if $Na^+$ is translocated towards the periplasm, MrpD LysTMH7/GluTMH5 pair will become neutral, as $Na^+$ charge has been removed and the proton from LysTMH7 can go back to GluTMH5. The absence of positive charge on LysTMH7 and the corresponding change in charge balance will now allow for MrpD LysTMH8 to be protonated by LysTMH12. In its turn, the

absence of charge on MrpD LysTMH12 will allow for the MrpA LysTMH7/GluTMH5 pair to become neutral, and then for MrpA LysTMH8 to become protonated by MrpA LysTMH12 in a similar sequence of events, with the system arriving to the state depicted in *Figure 5b*. The system is now reset, and the arrival of another $Na^+$ ion from the cytoplasm to the coupling site will set events in motion leading to state in *Figure 1a*, accompanied by the translocation of two protons into the cytoplasm. Here we discussed how $Na^+$ ion would drive proton translocation, but of course the reverse cycle is also applicable, and so proton translocation into the cell, driven by $\Delta\Psi$, would result in $Na^+$ being pumped out.

We believe that this mechanism explains all the properties of Mrp-catalysed antiport on the basis of prominent structural features. The ability to couple the translocation of two $H^+$ inside the cell in exchange for one $Na^+$ moving outside is achieved by the inter-MrpA$^N$-MrpD coupling point. A similar coupling principle, but with a cation binding site instead of LysTMH12 and an opposing directionality, is employed at the MrpD-MrpG interface. Coordinating/amplifying conformational interactions are possible because a break in TMH7 directly contacts a break in TMH8, while the loop from the TMH12 break directly contacts GluTMH5 from neighbouring APLS. Helix HL connects TMH7s on the cytoplasmic side (*Figure 2d*) and additionally, striking β-hairpin (βH) elements connect APLS on the periplasmic side both in Mrp and in complex I. Links to βH both of TMH5 (D157*a* and D128*d*) and TMH8 (D104*a* and D75*d*) are critical for activity (*Supplementary file 1*). Finally, a striking tilt of TM helices in the $Na^+$-translocating part of the complex as compared to the $H^+$-translocating domain (*Figure 2d*) might have a role in coupling if helices change the tilt during the catalytic cycle.

This two-APLS unit can be elegantly extended to three (*Sazanov, 2015*) or four (*Chadwick et al., 2018*) APLS in complex I-like enzymes or shortened to one in MBH (or FHL-1 or Ech enzymes [*Efremov and Sazanov, 2012*]), with the same mechanism applicable. The main difference to Mrp would be that in complex I (or membrane-bound hydrogenases [*Efremov and Sazanov, 2012*]) instead of a positive charge of $Na^+$, the coupling is achieved with a net negative charge being produced as two protons are abstracted during electron transfer and reduction of quinone to quinol or reduction of protons to hydrogen gas. The linking of three APLS units in complex I allows for the translocation of three protons in a similar to Mrp mechanism, while the fourth proton is likely translocated via the E-channel, as depicted in *Figure 5c*. In complex I the additional APLS subunit Nqo13/NuoM is unique as it has a conserved GluTMH12 instead of a lysine, which may allow this subunit to operate in an anti-phase with the other two APLS. Such a feature may be important to prevent excessive build-up of electrostatic imbalance in the membrane domain of complex I during the catalytic cycle. The E-channel structurally coincides with the exit path for $Na^+$ (*Figure 3b*) while the entry part is not homologous and is replaced in complex I with many glutamates (hence the E-channel) leading on towards the quinone-binding site as a putative proton 'relay'.

In MBH the redox [NiFe]-hydrogenase module is attached to a single APLS unit from the opposite site as compared to complex I (*Figure 3a*). As noted above, the additional $H^+$ pathway proposed for MBH (*Yu et al., 2018*) is very unlikely from structural considerations, and in any case it would lead to a futile cycle of protons coming in via MbhH and exiting via this additional pathway. It was discussed previously that the hydrogenase module of MBH evolves $H_2$ and generates a proton gradient, whereas the Mrp module transforms it into a $Na^+$ gradient (*McTernan et al., 2014*). We suggest that the $Na^+$ pathway in MBH is the same as we propose for Mrp, while the energy of the redox reaction (which is quite small for MBH with $\Delta E = 60$ mV [*Yu et al., 2018*]) is used to boost the translocation of a proton into the cell via MbhH, a standard APLS with all the key residues conserved (*Figure 3c*). The reversal of the redox module compared to complex I would then make complete sense because the redox energy would be used in MBH to drive proton translocation in the opposite direction as compared to complex I or FHL-type hydrogenases (*Efremov and Sazanov, 2012*), and so from the electrostatics point of view the charge would need to be delivered to APLS from the opposite side. Thus, instead of two APLS units driving $Na^+$ in Mrp, in MBH one APLS unit supported by the redox reaction would drive $Na^+$ in a similar to Mrp mechanism (*Figure 5d*).

In conclusion, the Mrp structure revealed basic operating principles of this ancient antiport system, which forms the basis of a huge variety of modern redox proton and sodium pumps. The mechanism that we propose, based on electrostatics and supported by conformational interactions, is likely to be applicable to all members of this group.

# Materials and methods

## Key resources table

| Reagent type (species) or resource | Designation | Source or reference | Identifiers | Additional information |
|---|---|---|---|---|
| Gene (*Thermus thermophilus*) | HB8, ATCC 27634, DSM 579 | Uniprot | Nqo12: Q56227<br>Nqo13: Q56228<br>Nqo14: Q56229 | Multiple sequence alignments |
| Gene (*Pyrococcus furiosus*) | COM1, DSM 3638 | pdb FASTA sequence | 6CFW | Multiple sequence alignments |
| Gene (*Bacillus pseudofirmus*) | Strain OF4 | Uniprot | MrpA: Q9RGZ5<br>MrpB: Q9RGZ4<br>MrpC: Q9RGZ3<br>MrpD: Q9RGZ2<br>MrpE: Q9RGZ1<br>MrpF: Q9RGZ0<br>MrpG: Q9RGY9 | Multiple sequence alignments |
| Gene (*Bacillus subtilis*) | 168 | Uniprot | MrpA: Q9K2S2<br>MrpB: O05259<br>MrpC: O05260<br>MrpD: O05229<br>MrpE: Q7WY60<br>MrpF: O05228<br>MrpG: O05227 | Multiple sequence alignments |
| Gene (*Staphylococcus aureus*) | 1280 | Uniprot | MrpA: Q9ZNG6<br>MrpB: P60678<br>MrpC: P60682<br>MrpD: P60686<br>MrpE: P60690<br>MrpF: P60694<br>MrpG: P60698 | Multiple sequence alignments |
| Gene (*Thermosynechococcus elongatus*) | *BP-1* | Uniprot | NdhF1: Q8DKX9<br>NdhD1: Q8DKY0 | Multiiple sequence alignments |
| Sequence-based reagent | MrpF D35L | This paper | PCR primers | <u>Forward</u>: CGCTTATTTTTTACTATATA TGTTGAAAAAAAATGAAAC<br><u>Reverse</u>: GCAAGCGTAATGCCCATCG CTAAGAGCGCGATAATACGATCCG<br><u>Forward</u>: CGGATCGTATTATCGCGCT CTTAGCGATGGGCATTACGCTTGC<br><u>Reverse</u>: ATATAGTAAAAAATAAGCG |
| Sequence-based reagent | MrpG S72W | This paper | PCR primers | <u>Forward</u>: CGCTTATTTTTTACTATAT ATGTTGAAAAAAAATGAAAC<br><u>Reverse</u>: CACGATGCCAAGCAATAGACG CCAGTTGAAATGGTTATTTTCAATG<br><u>Forward</u>: CATTGAAAATAACCATTTCAA CTGGCGTCTATTGCTTGGCATCGTG<br><u>Reverse</u>: ATATAGTAAAAAATAAGCG |
| Sequence-based reagent | MrpE L41W | This paper | PCR primers | <u>Forward</u>: CGCTTATTTTTTACTATATAT GTTGAAAAAAAATGAAAC<br><u>Reverse</u>: GCGCGAATGGAAAAAGCGACGCCAT ATAAAAAGAATAAACAGCCCGATCATGTAC<br><u>Forward</u>: GTACATGATCGGGCTGTTTATTCTTT TTATATGGCGTCGCTTTTTCCATTCGCGC<br><u>Reverse</u>: ATATAGTAAAAAATAAGCG |
| Strain, strain background (*Escherichia coli*) | KNabc | doi:10.1073/pnas.84.9.2615 | | Expression and assay strain |

*Continued on next page*

*Continued*

| Reagent type (species) or resource | Designation | Source or reference | Identifiers | Additional information |
|---|---|---|---|---|
| Transfected construct (*Anoxybacillus flavithermus*) | DSM 21510/WK1 | Other | Uniprot MrpA: B7GL84 MrpB: B7GL83 MrpC: B7GL82 MrpD: B7GL98 MrpE: B7GL97 MrpF: B7GL96 MrpG: B7GIG3 | Prof. Masahiro Ito (Graduate School of Life Sciences, Toyo University, Japan) |
| Software, algorithm | SerialEM | doi:10.1016/j.jsb.2005.07.007 | | Data Acquisition Software for a variety of data from electron microscopes |
| Software, algorithm | Relion | doi:10.1016/j.jsb.2012.09.006 | | Cryo-EM processing software |
| Software, algorithm | CTFFIND4 | doi:10.1016/j.jsb.2015.08.008 | | Defocus estimation software |
| Software, algorithm | Gctf | doi:10.1016/j.jsb.2015.11.003 | | Per-particle CTF estimation software |
| Software, algorithm | USCF Chimera | doi:10.1002/jcc.20084 | | Visualisation software of molecular structures and cryo-EM maps |
| Software, algorithm | Coot | doi:10.1107/S0907444904019158 | | Software for model building |
| Software, algorithm | PHENIX | doi:10.1107/S0907444909052925 | | Structure refinement software |
| Software, algorithm | MotionCor2 | doi:10.1038/nmeth.4193 | | Whole frame image motion correction software |

## Expression and purification of Mrp

Plasmid DNA, termed AF_Mrp, encoding Mrp from thermophilic *Anoxybacillus flavithermus* WK1 with a C-terminal His-tag on MrpG was kindly provided by Prof. Masahiro Ito (Graduate School of Life Sciences, Toyo University, Ouragun, Gunma 374–0193, Japan). AF_Mrp was expressed in 50 litres of LBK media (1% tryptone, 0.5% yeast extract, 83 mM KCl, pH 7.5) supplemented with 100 µg/mL ampicillin, 25 µg/mL kanamycin and antifoam using a fermenter. The bacterial culture was grown for 16–17 hr at 37°C, while keeping at constant pH of 7.3. The agitation was adjusted to keep the dissolved oxygen concentration between 2% - 10%. The next day, the cells were harvested by centrifugation (5000 x g for 30 min at 4°C) and the cell pellets were stored at −80°C.

## Preparing membranes from *E. coli* KNabc

Frozen cell pellets were thawed in ice-cold water and resuspended in 20 mM HEPES-KOH pH 7.0, 5 mM MgCl$_2$ and 10% glycerol. The cells were homogenized twice at 30,000 psi, using a high-pressure cell disruption Constant System pressure cell TS 1.1. DNases I (0.3 mg/ml) and proteinase inhibitor cocktail (5 tablets of EDTA-free Complete Ultra inhibitor [Roche]) were added to the lysate. The lysate was clarified by centrifugation at 26,000 x g for 30 min at 4°C. Membranes were obtained by ultracentrifugation of the supernatant at 180,000 x g for 2 hr at 4°C. Pelleted membrane fractions were resuspended in 20 mM HEPES-KOH pH 7.0, 5 mM MgCl$_2$, 20% glycerol and stored at −80°C.

## Membrane solubilisation

Membranes (10 mg/ml) were thawed in ice-cold water and solubilized by incubation for 1 hr at 4°C in 20 mM HEPES-KOH pH 7.0, 5 mM MgCl$_2$, 10% glycerol to which 0.3 M KCl and 1% (w/v) Lauryl Maltose Neopentyl Glycol (LMNG) or n-Dodecyl β-D-maltoside (DDM) had been added. After ultra-centrifugation at 180,000 x g for 15 min, the supernatant was diluted two-fold with 20 mM HEPES-KOH pH 7.0, 5 mM MgCl$_2$, 10% glycerol, 0.3 M KCl and imidazole was added up to a concentration of 20 mM.

## Purification

Solubilised membranes were loaded onto a 5 ml TALON (GE Healthcare) column, previously equilibrated with five column volumes (CV) of 20 mM HEPES-KOH pH 7.0, 5 mM MgCl$_2$, 10% glycerol, 0.3 M KCl, 20 mM imidazole and 0.05% LMNG or 0.05% DDM. The column was washed with 15 CV of equilibration buffer and the protein was eluted with 10 CV 20 mM HEPES-KOH pH 7.0, 5 mM MgCl$_2$, 10% glycerol, 0.3 M KCl, 200 mM imidazole and 0.05% LMNG or 0.05% DDM. Fractions containing Mrp were pooled and applied onto a Superose 6 10/300 GL (GE Healthcare) previously equilibrated with 2 CV of 20 mM Bis-Tris pH 6.0, 5 mM MgCl$_2$, 0.15 M KCl and 0.05% LMNG or 0.05% DDM, and eluted isocratically. The dimer eluted at around 12.5 ml and was diluted to 0.16 mg ml$^{-1}$ for grid preparation. Purification of the Mrp complex was performed more than five times, each time from a different batch of cells, and all attempts of replication were successful.

## Electron microscopy

Copper grids (Quantifoil mesh 300, R 0.6/1) were covered with a 1.2 nm thin layer of continuous carbon. Grids were glow discharged in air at 30 mA for 5 s. 3 µL of protein sample were applied to the grids, blotted for 7 s at 4℃ and 100% humidity and quickly plunged into liquid ethane using a FEI Vitrobot IV. Grids were stored in liquid nitrogen. Images were collected at 35° tilt using a 300 kV Titan Krios electron microscope equipped with a Gatan K3 camera and an energy filter set to a slit width of 20 eV at the Institute of Science and Technology Austria. Micrographs were collected with the FEI EPU package for the LMNG dataset or SerialEM for the DDM dataset, at a nominal magnification of 105,000 x, resulting in a calibrated physical pixel size of 0.84 Å per pixel. Defocus values varied from 0.6 µm to 2.3 µm. A total dose of 85 e$^-$/Å$^2$ was fractionated into 85 frames for the LMNG-dataset. A total dose of 90 e$^-$/Å$^2$ was fractionated into 88 frames for the DDM-dataset.

## Image processing of the LMNG-dataset

760 and 3126 movies were collected in normal- and super-resolution respectively. Processing was done in Relion 3.0.7 (*Scheres, 2012*). Movie frames were motion-corrected, dose-weighted and super-resolution images were binned two-fold using MotionCor2 (*Zheng et al., 2017*). Contrast transfer function (CTF) parameters were determined for aligned micrographs using CTFFIND4 (*Rohou and Grigorieff, 2015*). After a manual inspection of the Thon rings, bad micrographs showing ice rings were excluded from further analysis, yielding 755 good micrographs for normal resolution and 3096 for super resolution movies. Auto-picking with 3D references, which were the 3D auto-refined dimer map from the DDM dataset in C2 symmetry low-pass-filtered to 30 Å, resulted in 198565 particles for normal resolution and 693504 particles for super resolution micrographs. The coordinates of the particles were then used for a per-particle CTF estimation using Gctf (*Zhang, 2016*). Star files from normal and super-resolution micrographs were merged after Gctf estimation. At this stage and at all stages during the entire processing of both LMNG and DDM datasets, when particles were re-extracted after classification or refinement, the duplicates were removed using 100 Å minimum inter-particle distance. 2D classification was attempted but it did not improve the results. 3D classification of all picked particles, extracted in a 256 pixel box (down-sampled to 1.68 Å pixel) was carried out in C1 symmetry with a 3D auto-refined dimer map from the DDM dataset as initial reference, filtered to 30 Å. This resulted in four good classes with 606671 particles. The best class was selected yielding 264961 particles. Particles were re-extracted in a 512 pixel box (0.84 Å pixel). An initial 3D auto-refinement in C2, with local angular searches and a 3D auto-refined dimer map from the DDM dataset filtered to 30 Å as a reference resulted in a map with an overall resolution of 5.5 Å.

## Monomer

The resolution of the dimer map was limited due to variable angle between the monomers, resulting in the loss of true C2 symmetry for the entire particle pool. The particles were therefore symmetry-expanded according to the C2 point group, meaning that the particle number was enlarged twice because each dimer particle produced two monomer particles. Particles were re-extracted with re-centring on a monomer in a 512 pixel box (0.84 Å pixel). Removal of duplicates during re-extraction resulted in the loss of a few particles, which were probably coming from two neighboring dimers, and so the final number of particles is slightly less than double. 3D classification of these monomers,

with local angular searches and a monomer map (excised in Chimera from the auto-refined dimer map and filtered to 8 Å) as a reference resulted in two good classes, with combined 285688 particles. These particles were re-extracted in a 512 pixel box (0.84 Å pixel) for masked 3D auto-refinement in C1, with local searches and the same monomer map as during 3D classification as a reference, which resulted in a map with an overall resolution of 3.16 Å. After post processing and polishing, 3D auto-refinement was repeated with local angular searches and a monomer map as a reference, resulting in a map with an overall resolution of 3.05 Å. The final resolution after post processing was 2.98 Å.

### Dimer

The processing of the dimer was done in the same way as the processing of the monomer up to the first 3D classification. After 3D classification the best class was selected, extracted in a 256 pixel box (down-sampled to 1.68 Å pixel) and another round of 3D classification was done in C1 symmetry with global searches and a dimer map (a 3D auto-refined dimer map from the DDM dataset filtered to 30 Å) as initial reference. 3D classification resulted in one best class with 59328 particles. Masked 3D auto-refinement of this class, re-extracted in a 512 pixel box (0.84 Å pixel) was conducted in C2 symmetry with a map of this class filtered to 8 Å as a reference, resulting in a map with an overall resolution of 3.7 Å. The final resolution after post processing was 3.74 Å. Other dimer classes could be refined to resolutions of about 4 Å and differed only by the angle between the monomers.

### Image processing of the DDM-dataset

1544 movies were collected in super-resolution mode. Processing was done in Relion 3.0.2. Movie frames were motion corrected, dose weighted and binned two-fold using MotionCor2. CTF parameters were determined for each micrograph from non-dose-weighted, aligned images using CTFFIND4. The data was manually examined and micrographs showing poor power spectra, large portions of carbon or extensive ice-contaminations were excluded, yielding 1255 good micrographs. Auto-picking with 3D references filtered to 30 Å resulted in 226371 particles. The 3D references came from a 3D auto-refined map that was generated in a previous low-resolution test data set. The coordinates of the particles were then used for a per-particle estimation using Gctf. Particles were extracted in a 256 pixel box (down-sampled to 1.68 Å pixel). 2D classification resulted in four good classes with 150138 particles. 3D classification was carried out in C1 symmetry using as initial reference a 30 Å low-pass-filtered map that was generated in C2 symmetry using initial model generation tool in Relion.

### Monomer

After 3D classification two good classes were selected, resulting in 140351 particles. The particles were then symmetry-expanded according to the C2 point group, re-extracted with re-centring on a monomer in a 512 pixel box (0.84 Å pixel) and duplicates were removed, resulting in 272878 particles. 3D classification without a mask in C1 symmetry, a monomer map (excised in Chimera from the best 3D class dimer map and filtered to 8 Å) as a reference and local searches resulted in one good class with 83340 particles. Another masked 3D auto-refinement with particles re-extracted in a 512 pixel box (0.84 Å pixel), starting with local searches in C1 symmetry, followed by post-processing resulted in a map with a resolution of 3.7 Å. The map revealed that helices were left-handed (50:50 chance of that since the initial dimer model was generated de novo) and so this map and all further reference maps for monomers and dimers had their hand inverted for further processing. Particle-polishing and 3D auto-refinement improved the resolution to 3.41 Å.

### Dimer

After first 3D classification, two good classes with 140351 particles were selected. Several rounds of 3D classification were performed with the reference model in the correct hand (initial model with inverted hand and filtered to 30 Å), which resulted in one good class with 89240 particles. Further 3D classification with global searches in C1 symmetry resulted in three classes with 10125, 21450 and 25395 particles per class. Masked 3D auto-refinement with particles extracted in a 512 pixel box (0.84 Å pixel), in C2 symmetry and local searches was performed with each class. After post-processing, this resulted in maps with resolutions of 7.5 Å, 8.0 Å and 4.3 Å.

## Atomic model building

The initial model was generated using the cryo-EM structure of the MBH complex (*Yu et al., 2018*) and the crystal structure of complex I from *Thermus thermophilus* (*Baradaran et al., 2013*). Homology models were created for all subunits of Mrp with Phyre2 server (*Kelley et al., 2015*) using chain T from *Thermus thermophilus* complex I as a template for the N-terminal part of MrpA and MbhD together with MbhE for the C-terminal part of MrpA. MbhF, MbhG, MbhH, MbhA, MbhB, MbhC from MBH and were used as templates for MrpB, MrpC, MrpD, MrpE, MrpF and MrpG, respectively. The homology models were fit into our cryo-EM map using USCF Chimera (*Pettersen et al., 2004*). Morphing was used to adjust the model to fit the cryo-EM map using PHENIX software (*Adams et al., 2010*). The model was then manually corrected using Coot (*Emsley and Cowtan, 2004*) and refined against the cryo-EM map in real space using PHENIX with our protocol for cryo-EM structure refinement which allows electron radiation-damaged carboxyl side-chains to acquire high B-factors, so they don't distort the backbone (*Letts et al., 2019*). Densities for several lipids could be detected. Based on the appearance of their density and the prevalence of phosphatidylethanolamine among *E. coli* lipids, phosphatidylethanolamine was modelled into these densities.

The initial model was built into the DDM-dataset monomer density and then extended and completed in the LMNG-dataset monomer density. For accurate modelling of water molecules, particularly to avoid false positives, we filtered the LMNG monomer map by local resolution in Relion and resampled it at 0.5 Å per pixel (akin to the water modelling procedure in phenix.douse). After this procedure, water molecules displayed strong signals (~2 rmsd), had nearly spherical densities, were not clashing with other atoms and participated in hydrogen bonds, which are all strongly indicative of real water molecules. This allowed automatic placement of water molecules in COOT, which were then all checked and corrected manually, to leave only waters with clear density and fulfilling geometry criteria. $K^+$ ions were placed on the basis of density and a coordination pattern by nitrogen atoms from histidines and oxygen atoms from waters or serines and glutamine. The average coordination distance was about 2.8–3.0 Å, consistent with known values for $K^+$ (*Harding, 2002*). Mg was present in the crystallisation buffer but was excluded as a candidate for bound ions since it should have stronger density and is normally coordinated by at least several negatively charged residues, with average coordination distances of about 2.2 Å (*Zheng et al., 2008*). One of K ions (#1, to the left in *Figure 2—figure supplement 1e,f*) is found in the identical positions in LMNG and DDM maps, while another one (#2, to the right in *Figure 2—figure supplement 1e,f*) is shifted by about 3 Å. However, ion #2 is still located within the same cavity and is coordinated by the same residues in both cases (H40*c*, H37*c* and N702*a*). The difference is probably due to partly disordered/disturbed DDM structure. Therefore we conclude that DDM density supports the notion that cavity is able to coordinate two $K^+$ ions, but for a detailed discussion we use $K^+$ ions from the higher resolution and more complete LMNG model.

For the final refinement of the dimer, two monomer densities at 3.0 Å resolution were combined in Chimera after their fit into the 3.7 Å resolution dimer density. Two monomer models were fit into this composite map and refined in one final round.

## Introduction of site-directed mutations

Gibson assembly (*Gibson et al., 2009*) was used for the construction of plasmids with point mutations in AF_Mrp (MrpE L41W, MrpG S72W and MrpF D35L). The point mutations in MrpE and MrpG were chosen to disrupt the Mrp dimer. The point mutation in MrpF was chosen due to the putative involvement of this residue in $Na^+$ translocation. Each mutated plasmid was generated by combining two big DNA fragments of similar size, which were produced by two independent PCRs using AF_Mrp as template by means of Gibson assembly. The Gibson assembly method requires that the DNA fragments have ~20 base pairs overlaps with the adjacent fragment. The overlaps were added to the ends of the fragments by means of long (~40 base pairs) primers, which also contained the point mutation. The sequence of the mutated plasmids was confirmed by sequencing.

## Preparation of everted membrane vesicles

Everted membrane vesicles (EMV) were prepared as previously described (*Ambudkar et al., 1984*), with some changes. *E. coli* KNabc cells were transformed with the respective plasmid and grown in LBK medium containing the respective antibiotics for 16 hr at 37˚C. The cells were harvested at 4000

x g and the pellet was washed two times with 10 mM Bis-Tris-Propane-Sulfate pH 7.5, 5 mM $MgCl_2$, 140 mM choline chloride and 10% glycerol and resuspended in the same buffer. Constant System pressure cell was used to prepare EMV by passing the resuspended cells through the cell disruptor for a single time at 10,000 psi. The cell suspension was centrifuged at 36,000 x g for 15 min followed by a centrifugation at 180,000 x g for 1.5 hr. The same procedure was done for the control, which were non-transformed KNabc cells. The EMV were suspended at 20 mg/ml and stored at $-80°C$.

## Antiport assay

$Na^+/H^+$ antiport activity assay was performed as previously described (*Morino et al., 2008*; *Swartz et al., 2007*). 66 µg of EMV were suspended in 2 ml 10 mM Bis-Tris-Propane-Sulfate, 140 mM choline chloride, 5 mM $MgCl_2$, 1 µM acridine orange at pH 7.5, pH 8.5 or pH 9.5. Measurements were conducted with excitation and emission at 420 and 500 nm, respectively, using a Spectramax M2e Plate and Cuvette reader. Tris-succinate at pH 7.5, pH 8.5 or pH 9.5 was added to a final concentration of 2.5 mM to initiate respiration. After fluorescence quenching of acridine orange, a steady-state was reached and NaCl or KCl was added to a final concentration of 2.5 mM. The $Na^+$ dependent fluorescence dequenching is indicative for the $Na^+/H^+$ antiport activity. $NH_4Cl$ was added to a concentration of 10 mM to bring the fluorescence back to baseline. Assays were conducted in triplicates with three independent membrane preparations.

## Growth experiments

Growth experiments were conducted as previously described (*Kosono et al., 2006*). *E. coli* KNabc cells were transformed with the respective plasmid and incubated for 16 hr at 37°C in LBK medium containing 0–1200 mM NaCl. KNabc cells lack the three main sodium proton antiporters and can withstand a salt concentration of up to 200–300 mM. A functional Mrp antiporter complements the inability of KNabc cells to grow at higher salt concentrations.

# Acknowledgements

This research was supported by the Scientific Service Units (SSU) of IST Austria through resources provided by the Electron Microscopy Facility (EMF), the Life Science Facility (LSF) and the IST high-performance computing cluster. We thank Dr Victor-Valentin Hodirnau and Daniel Johann Gütl from IST Austria for assistance with collecting cryo-EM data. We thank Prof. Masahiro Ito (Graduate School of Life Sciences, Toyo University, Japan) for a kind provision of plasmid DNA encoding Mrp from *A. flavithermus* WK1. JS is a recipient of a DOC Fellowship of the Austrian Academy of Sciences at the Institute of Science and Technology, Austria.

# Additional information

### Funding

| Funder | Grant reference number | Author |
| --- | --- | --- |
| Austrian Academy of Sciences | DOC fellowship | Julia Steiner |

The funders had no role in study design, data collection and interpretation, or the decision to submit the work for publication.

### Author contributions

Julia Steiner, Formal analysis, Funding acquisition, Investigation, Visualization, Methodology, Writing - review and editing; Leonid Sazanov, Conceptualization, Resources, Supervision, Funding acquisition, Validation, Investigation, Visualization, Methodology, Writing - original draft, Project administration, Writing - review and editing

### Author ORCIDs

Leonid Sazanov https://orcid.org/0000-0002-0977-7989

Decision letter and Author response
Decision letter https://doi.org/10.7554/eLife.59407.sa1
Author response https://doi.org/10.7554/eLife.59407.sa2

## Additional files

### Supplementary files

• Supplementary file 1. Supplementary tables 1– 3.

• Transparent reporting form

### Data availability

Structure of the Mrp dimer is deposited in PDB with PDB ID 6Z16, with corresponding cryo-EM density maps in EMDB (EMD-11027).

The following datasets were generated:

| Author(s) | Year | Dataset title | Dataset URL | Database and Identifier |
|---|---|---|---|---|
| Steiner J, Sazanov LA | 2020 | Structure of the Mrp antiporter complex | https://www.rcsb.org/structure/6Z16 | RCSB Protein Data Bank, ID6Z16 |
| Steiner J, Sazanov LA | 2020 | Structure of the Mrp antiporter complex | https://www.ebi.ac.uk/pdbe/entry/emdb/EMD-11027 | Electron Microscopy Data Bank, EMD-11027 |

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
