## [Decision Letter]

**Acceptance summary:**

Multiple resistance and pH adaptation proteins are a set of multi-subunit antiporters that exchange sodium or potassium ions for protons. The authors have succeeded in determining an atomic model for the structure for a representative member of this protein family using cryo-EM. The 100-transmembrane helix dimer reported here is a tour de force in membrane protein structure determination and enables elucidation of a mechanism where the coupling between sodium and proton translocation is facilitated by a series of electrostatic interactions between a cation and key charged residues.

**Decision letter after peer review:**

Thank you for submitting your article "Structure and mechanism of the Mrp complex, an ancient cation/proton antiporter" for consideration by *eLife*. Your article has been reviewed by three peer reviewers, and the evaluation has been overseen by Sriram Subramaniam as Reviewing Editor and Olga Boudker as the Senior Editor. The following individuals involved in review of your submission have agreed to reveal their identity: Joel Meyerson (Reviewer #1); Doreen Matthies (Reviewer #2).

The reviewers have discussed the reviews with one another and the Reviewing Editor has drafted this decision to help you prepare a revised submission.

Summary:

Multiple resistance and pH adaptation (Mrp) proteins are multi-subunit antiporters that exchange Na^+^ or K^+^ for H^+^ and represent an ancestor of important redox driven proton pumps such as Complex I. At high extracellular pH Mrp is hypothesized to translocate 2H^+^ / 1Na^+^, and it is proposed to have a large surface area to allow for the few available external protons to be gathered for translocation into the cell. However, the mechanism of coupling in this family is unknown. This study succeeds in resolving a structure Mrp and provides a comprehensive analysis and comparative analysis with MBH and Complex I.

Essential revisions:

Please address all of the points below in the revised manuscript:

• How did the authors discriminate between water molecules, K^+^ ions, and noise during modeling? This is an important part of the analysis and details should be given in the manuscript.

• The authors provided a map and model which appears to be the Mrp-LMNG structure. The structure of Mrp in DDM was also used for analysis, including for localizing K^+^ ions. This map and model should be made available.

• Figure 2—figure supplement 1F: In panels E and F (DDM and LMNG maps) the proposed K^+^ ions appear in different positions. Can the authors comment on this?

• Water molecules are modeled in Figure 2—figure supplement 1 but they do not appear to be present in the PDB file included with the submission.

• In Figure 2A and in the manuscript the authors talk about membrane thinning near the dimerization area of the complex based on hydrophobicity analysis of the surface of the structure. The detergent belt around the protein in the map can be a good indication of the location of the membrane and its thickness and I encourage the authors to analyze their unmasked or lower resolution maps to either support their claim or not. Also if any lipid or detergent densities are visible in the high resolution maps near that location and the head groups on the outer side are actually shifted, I'd be more convinced of this claim than just simply based on hydrophobicity. Also, the final dimer map has a 20 degree tilt between the monomers. I understand that this small subset was identified during 3D classification and resulted in the highest resolution map but more than 50% of the dimers seem to show tilts of 4-9 degrees only. This raises the question on how representative this particular dimer structure is of this complex.

• Two densities have been assigned to K^+^ and interpreted as Na^+^ binding sites. These densities have also been observed in the 3.41 A DDM monomer map. What are the arguments that this is indeed K^+^ and not stabilizing Mg ions? The buffer contains 5 mM MgCl2. These specific densities assigned to K^+^ in Figure 2—figure supplement 1E and F (displaying the DDM and LMNG maps) also look like they are at different locations. Figure 4—figure supplement 1C shows Na binding sites. This interpretation should be described very cautiously or supported by molecular simulation studies.

• The manuscript can be made more accessible to readers if the authors would (i) define their abbreviations (NDH, MBH, APLS….) when first used in the text, (ii) use the same notations to refer to residues either by subunit and residue number or by the transmembrane helix/subunit, (iii) use the same nomenclature as given by labels in figures to allow readers to readily identify which residues are being referred to, (iv) edit the text to eliminate typos (e.g. subsection “Antiport mechanism”, third paragraph Gly or Glu, and subsection “Cation (Na^+^ or K^+^) translocation pathway”, second paragraph H42g or H32g?).

---

## [Author Response]

Essential revisions:Please address all of the points below in the revised manuscript:• How did the authors discriminate between water molecules, K^+^ ions, and noise during modeling? This is an important part of the analysis and details should be given in the manuscript.

We added this information to Materials and methods:

“For accurate modelling of water molecules, particularly to avoid false positives, we filtered the LMNG monomer map by local resolution in Relion and resampled it at 0.5 Å per pixel (akin to the water modelling procedure in phenix.douse). […] Mg was present in the purification buffer but was excluded as a candidate for bound ions since it should have stronger density and is normally coordinated by at least several negatively charged residues, with average coordination distances of about 2.2 Å (Zheng et al., 2008).”

• The authors provided a map and model which appears to be the Mrp-LMNG structure. The structure of Mrp in DDM was also used for analysis, including for localizing K^+^ ions. This map and model should be made available.

We have included with the revised manuscript the DDM monomer map and the final monomer model fitted in this map. Included K^+^ ions were fitted into this DDM map, as used for Figure 2 —figure supplement 1. We do not think it is informative to provide the initial DDM model, as it was severely incomplete, as described in the manuscript.

• Figure 2—figure supplement 1: In panels E and F (DDM and LMNG maps) the proposed K^+^ ions appear in different positions. Can the authors comment on this?

Indeed, one of the ions (#1, to the left in Figure 2—figure supplement 1E, F) is found in the identical positions in LMNG and DDM maps, while another one (#2, to the right in Figure 2—figure supplement 1E, F) is shifted by about 3 Å. However, ion #2 is still located within the same cavity and is coordinated by the same residues in both cases (H40c, H37c and N702a). The difference is probably due to partly disordered/disturbed DDM structure. Therefore, we conclude that the DDM density supports the notion that the cavity is able to coordinate two K^+^ ions, but for a detailed discussion we used K^+^ ions from the higher resolution and more complete LMNG model.

We have now added this information into the Materials and methods section.

• Water molecules are modeled in Figure 2—figure supplement 1 but they do not appear to be present in the PDB file included with the submission.

With the revised manuscript we have included the LMNG monomer map filtered by local resolution in Relion and then re-sampled at 0.5 Å, as described above for modeling of waters. The monomer model with waters, refined in this map, is also included. Since due to the still limited resolution we used these experimental waters only for the comparison with predictions from Dowser, and use Dowser waters for the analysis of proton translocation pathways, we would prefer not to deposit these files in the PDB. The PDB-deposited dimer model is more representative of the physiological unit consisting of intertwined monomers.

• In Figure 2A and in the manuscript the authors talk about membrane thinning near the dimerization area of the complex based on hydrophobicity analysis of the surface of the structure. The detergent belt around the protein in the map can be a good indication of the location of the membrane and its thickness and I encourage the authors to analyze their unmasked or lower resolution maps to either support their claim or not. Also if any lipid or detergent densities are visible in the high resolution maps near that location and the head groups on the outer side are actually shifted, I'd be more convinced of this claim than just simply based on hydrophobicity.

We thank the reviewer for this suggestion – we have added Figure 2—figure supplement 1H), which shows the detergent belt density in the low-resolution (~ 7 Å) map of LMNG dimer. The belt is indeed much thinner in the dimerization area, supporting our proposal. There are no well-resolved lipids in this area, unfortunately. We also think that Figures 2A and Figure 2—figure supplement 1G illustrate quite strongly that not only the hydrophobic exposed belt is thinner at the dimer interface, there is simply not enough protein to fully traverse the intact membrane – due to the very short TMH1/2 from MrpE subunit, the protein is only about 20 Å thick at the dimer interface. We now mention these facts in the subsection “Overall structure”.

Also, the final dimer map has a 20 degree tilt between the monomers. I understand that this small subset was identified during 3D classification and resulted in the highest resolution map but more than 50% of the dimers seem to show tilts of 4-9 degrees only. This raises the question on how representative this particular dimer structure is of this complex.

In the LMNG dataset about 55% of particles had angles between 4-9 degrees and about 45% between 17-20 degrees. In the DDM dataset all particles had angles between 20-28 degrees. We mentioned in the original text that the LMNG dimers were overall “flatter” than the DDM dimers and so may be closer to their native shape since the lipid bilayer is flat. We did not imply that they have to be absolutely flat in terms of the apparent angle between the monomers. Further considerations indicate that the ~20 degree dimers are probably closest to native, as in this case the exposed hydrophobic belt area is closer to being flat (Figure 2A). The impression of a relative tilt of monomers is exacerbated by the quite large hydrophilic cytoplasmic domains at the dimer interface. Furthermore, all dimers refining to the highest resolution had angles close to 20 degrees, suggesting that they are probably closest to their physiological state. As we also needed the highest resolution dimers to verify the contacts between the monomers, we used the best class (leading to 3.7 Å map, Figure 1—figure supplement 3) for further analysis. Comparisons of maps of various dimers did not reveal any specific differences in the overall structure, apart from different apparent angles between the monomers, suggesting that the discussed structure is a valid representative. We have added these considerations in the subsection “Structure determination”.

• Two densities have been assigned to K^+^ and interpreted as Na^+^ binding sites. These densities have also been observed in the 3.41 A DDM monomer map. What are the arguments that this is indeed K^+^ and not stabilizing Mg ions? The buffer contains 5 mM MgCl2. These specific densities assigned to K^+^ in Figure 2—figure supplement 1E and F (displaying the DDM and LMNG maps) also look like they are at different locations. Figure 4—figure supplement 1C shows Na binding sites. This interpretation should be described very cautiously or supported by molecular simulation studies.

We replied to all of these questions in detail above. Figure 4—figure supplement 1C legend is expanded to clarify that K^+^ ions are observed.

• The manuscript can be made more accessible to readers if the authors would (i) define their abbreviations (NDH, MBH, APLS….) when first used in the text, (ii) use the same notations to refer to residues either by subunit and residue number or by the transmembrane helix/subunit, (iii) use the same nomenclature as given by labels in figures to allow readers to readily identify which residues are being referred to, (iv) edit the text to eliminate typos (e.g. subsection “Antiport mechanism”, third paragraph Gly or Glu, and subsection “Cation (Na^+^ or K^+^) translocation pathway”, second paragraph H42g or H32g?).

APLS, NDH, Fpo, MBH have been now defined when used first time. We use LysTMH7/8/12 and GluTMH5 nomenclature only for key conserved and essential lysine and glutamate residues which define proton channels in the three homologous antiporter-like subunits (APLS) in complex I and related proteins, including Mrp. This is a standard nomenclature in the field and it makes it much easier to follow discussion of proton channels, as it applies to any of APLS. Otherwise one will have to list two-three unique residues for the description of each feature, as well as mentioning each time where a particular residue sits, making it difficult for discussion. On the detailed structural figures, on the other hand, it makes much more sense to label each particular residue with its own residue number. This has been a tradition in complex I field for a long time already, so we would prefer to use it here as well.

Typos were corrected, thanks.